# Therapeutic Inhalation of Hydrogen Gas for Alzheimer’s Disease Patients and Subsequent Long-Term Follow-Up as a Disease-Modifying Treatment: An Open Label Pilot Study

**DOI:** 10.3390/ph16030434

**Published:** 2023-03-13

**Authors:** Hirohisa Ono, Yoji Nishijima, Shigeo Ohta

**Affiliations:** 1Departments of Neurosurgery and Neurology, Nishijima Hospital, Ohoka, 2835-7, Numazu City 410-0022, Japan; 2Department of Neurology Medicine, Graduate School of Medicine, Juntendo University, 2-1-1 Hongo, Bunkyo-ku, Tokyo 113-8421, Japan

**Keywords:** ADAS-cog, Alzheimer’s disease, diffusion tensor imaging, neuronal integrity, hydrogen gas, disease-modifying treatment, multiple functions

## Abstract

(1) Background: Alzheimer’s disease (AD) is a progressive and fatal neurodegenerative disorder. Hydrogen gas (H_2_) is a therapeutic medical gas with multiple functions such as anti-oxidant, anti-inflammation, anti-cell death, and the stimulation of energy metabolism. To develop a disease-modifying treatment for AD through multifactorial mechanisms, an open label pilot study on H_2_ treatment was conducted. (2) Methods: Eight patients with AD inhaled 3% H_2_ gas for one hour twice daily for 6 months and then followed for 1 year without inhaling H_2_ gas. The patients were clinically assessed using the Alzheimer’s Disease Assessment Scale-cognitive subscale (ADAS-cog). To objectively assess the neuron integrity, diffusion tensor imaging (DTI) with advanced magnetic resonance imaging (MRI) was applied to neuron bundles passing through the hippocampus. (3) Results: The mean individual ADAS-cog change showed significant improvement after 6 months of H_2_ treatment (−4.1) vs. untreated patients (+2.6). As assessed by DTI, H_2_ treatment significantly improved the integrity of neurons passing through the hippocampus vs. the initial stage. The improvement by ADAS-cog and DTI assessments were maintained during the follow-up after 6 months (significantly) or 1 year (non-significantly). (4) Conclusions: This study suggests that H_2_ treatment not only relieves temporary symptoms, but also has disease-modifying effects, despite its limitations.

## 1. Introduction

Alzheimer’s disease (AD) is a progressive and fatal neurodegenerative disorder that causes impairments in cognition, memory, and behavior. Most AD drug developments to date have focused on targeting a single mechanism according to the conventional strategy [1]. Currently approved drugs for AD treatment including cholinesterase inhibitors (donepezil, rivastigmine, and galantamine) and an N-methyl-D-aspartic acid receptor antagonist (memantine) are symptomatic, but poorly improve the progression of the disease [2]. On the other hand, aging is the highest risk factor for AD, suggesting that multiple factors are involved in the etiology of AD in a complex manner [3]. In recent years, multifactorial mechanisms and multi-target strategies have been considered to develop disease-modifying drugs [4,5]. However, suitable combinations of two or three medicines may be difficult because of the difficulty in exploring even one effective medicine. Thus, it is necessary to explore the utility of a multiple function molecule as a disease modifier. 

The direct cause of AD is neuronal death, and it is impossible to revive dead cells by any treatments. However, in addition to being temporarily activated, it may be possible that surviving neurons could be improved in the integrity of neurons after AD onset on receiving a disease-modifying treatment [6]. A more ideal strategy would be to restore the integrity of surviving neurons with disease-modifying therapies, even after the onset of AD.

Molecular hydrogen (H_2_) is an inert molecule in the absence of a catalyst. It has long been believed that H_2_ has no biological function in mammalian cells. H_2_ first reported in 2007 as a therapeutic antioxidant by one of the present authors [7]. Subsequent extensive studies on model animals revealed that H_2_ exerts multiple functions such as anti-inflammatory, anti-cell death, and the stimulation of energy metabolism to exhibit efficacies against a variety of disease models [8,9]. H_2_ has the ability to cross the blood-brain barrier (BBB) by gaseous diffusion without a specific drug delivery system [7]. A number of animal experiments suggest the potential of H_2_ to improve neurodegenerative disorders [10]. Moreover, numerous small-scale clinical studies have indicated that H_2_ therapy provides the marked beneficial effects in a wide range of diseases. It is noteworthy that no adverse effects have been reported in human studies related to the administration of H_2_ therapy [11]. In fact, the inhalation of H_2_ gas has been approved as safe by a Phase I clinical trial [12]. 

Several reports have shown marked efficacies of H_2_ in dementia model animals [13]. In addition, clinical studies suggest that H_2_ is applicable for dementia. We showed that long-term drinking of H_2_-dissolved water (H_2_ water) improved the cognition of subjects with mild cognitive impairment (MCI) who carry the apoE4 genotype [14], who have a high risk of AD [15]. Moreover, we recently published a case report on advanced AD involving a patient who showed improved fecal incontinence by the continuous 2-year inhalation of H_2_ gas [16]. Thus, H_2_ has a strong potential as a multi-functional agent for improving AD.

In the present study, we aimed to identify a treatment that activates neurons after the onset of AD as a disease-modifier. As an objective assessment of neuronal integrity, the diffusion tensor imaging (DTI) method [17,18,19], with an advanced magnetic resonance imaging (MRI) technique, was applied in addition to a clinical evaluation [20]. Here, we suggest that the inhalation of H_2_ gas exhibits a therapeutic effect. In addition, follow-up over the subsequent one year without inhaling H_2_ gas suggests that this treatment not only relieves temporary symptoms, but also provides a disease-modifying effect. 

## 2. Results

### 2.1. Hydrogen Treatment Improved AD as Assessed by ADAS-cog

Eight patients and their family members agreed to participate in this open label study. Table 1 shows the background of the participants and controls (Table 1). 

Since the Alzheimer’s Disease Assessment Scale-cognitive subscale (ADAS-cog) is widely accepted as one of the most trustworthy methods, we used ADAS-cog as a clinical assessment [20]. In ADAS-cog, a lower value means improvement and a higher value means worse. The initial stage of the ADAS-cog of the participants ranged from 19 to 40. The patients continued to inhale 3% H_2_ gas, for 1 h, twice per day, for 6 months. Throughout the study, no adverse effect was noted. 

ADAS-cog changes from each initial stage were obtained at 3 and 6 months, and the follow-up periods for an additional 6 and 12 months. The patients did not inhale H_2_ gas during the follow-up period. Figure 1 shows the time-dependent profile of the mean of each change in ADAS-cog. 

ADAS-cog control data were obtained from the Nishijima Hospital database in which patients with AD were followed approximately every 6 months for more than 1.5 years. The mean value of the initial ADAS-cog in the control untreated group was 27.8 with ±3.53 standard deviations, which was similar to that of the H_2_-treated group, and the profile of the untreated group on each change in ADAS-cog was in good agreement with previous publications [21,22]. 

The mean value of ADAS-cog change in the H_2_-treated group tended to increase (worsened) after the first 3 months (Figure 1). The long-term usage of a facial mask may cause the stress during the inhalation of H_2_ gas. After the next 3 months, the ADAS-cog change in H_2_ group was markedly decreased (improved) compared with the initial stage, while the untreated controls were continuously worsened. The improvement after the next 3 months (totally 6 months) was significantly greater than in the untreated control (Figure 1).

Moreover, the improvement became noticeable in the follow-up after 6 months even without inhaling H_2_ gas, in which this improvement was significant vs. the untreated group (Figure 1). In further follow-up for 6 additional months, the trend of improvement continued. 

### 2.2. Improvement of Neurons by H_2_ Inhalation as Assessed by Diffusion Tensor Imaging 

The neuron bundles of cerebral white matter are highly directional. The water diffusivity is much higher along the direction of the alive bundle compared with other directions. The difference between diffusivity along and across bundles increases depending upon neuronal integrity and axonal density. Neuron bundles that passed through the entire hippocampus at five seed positions were visualized by three-dimensional diagrams, and a two-dimensional image containing the hippocampus was arranged into one image (Figure 2). 

Fractional anisotropy (FA) values reflect the density of active axons within bundles, with lower values corresponding to lower active axonal density. To evaluate the diffusivity of water along the neuron bundles as neuron integrity, diffusion tensor imaging (DTI) was conducted with different FA values at 0.1 and 0.2 for the entire hippocampal region [17,18,19]. The neuron tract size visualized at FA = 0.2 reflects the higher neuronal integrity, while the visualized neuron tract reflects the entire neurons at FA = 0.1, which is less sensitive to neuronal integrity. Figure 2 visualized the neuron bundles that passed through the entire hippocampus at five seed positions (Figure 2).

Figure 3 shows the representative DTI pictures of two patients observed with lateral and axial views, at the initial stage, 6 months post-treatment, and follow-up for the subsequent 6 and 12 months. The visualized bundles at FA = 0.2 were increased after 6 months of treatment, and maintained during the follow-up. It is notable that the visualized bundles changed not only in size, but also in shape, suggesting that the H_2_ treatment improved the integrity of the additional area, as shown by the arrowheads in Figure 3.

Values for each measurement variation should be normalized for more accurate quantitative analysis. The value of FA = 0.1 is less sensitive for neuron integrity. Thus, the value of FA = 0.2 was divided by the value of FA = 0.1, which was obtained at the same time, and corrected for variation between measurements for normalization. Therefore, the number of pixels in the corresponding neuron bundle tracts at FA = 0.2 was normalized by the number of pixels at FA = 0.1, and the mean value was obtained by averaging the 4 images from each patient from the right and left hemispheres from the lateral and axial views. Five AD patients who consented to the use of their DTI data for this study were recruited. The ADAS-cog mean and standard deviation values (27.6 ± 6.5) were similar to the H_2_ group (Table 1).

Figure 4 shows the time course of the means with the standard errors. The mean value of the neuron tracts significantly increased with 6-month H_2_ inhalation vs. the control (*** *p* = 0.001) and vs. the initial stage (^##^ *p* = 0.0036). After the 6-month follow-up, the improvement was significantly maintained vs. the initial stage (^#^
*p* = 0.011). In addition, the mean value after 12-month follow-up was higher than the initial value, despite there being no significance.

Taken together, the inhalation of H_2_ gas for 6 months improved AD, as assessed by clinical ADAS-cog and objective DTI. Moreover, at least after 6 months without H_2_ treatment, the improvement was maintained, suggesting that this treatment provides not only temporary relief, but also a disease-modifying effect.

## 3. Discussion

It is widely accepted that the accumulation of amyloid β protein induces neuronal death in AD [23]. Alternatively, oxidative stress plays important roles in AD and is considered as one of the central factors in the pathogenesis of AD [24]. Since oxidative stress affects the expression of numerous genes that regulate many pathological phenomena such as increased production of amyloid, modification of the tau protein, autophagy, and apoptosis, the antioxidant effect of H_2_ may be its most important neuroprotective property [11].

The pathogenesis of AD involves strong interactions with immunological mechanisms in the brain [25]. Inflammation apparently occurs in pathologically vulnerable regions of the AD brain [26].

Additionally, the down-regulation of energy metabolism has been implicated as one of the risks and/or causes of AD. Deficits in glucose availability and mitochondrial function are well-known hallmarks of brain aging and are particularly emphasized in neurodegenerative disorders such as AD [27], implicating that energy metabolism is involved in their progression. In fact, moderate exercise has been suggested to help delay the progression of AD [28]. 

Therefore, one strategy for AD therapy or drug development is to modulate oxidative stress, inflammation, and energy metabolism [29]. 

Recently, a number of randomized clinical studies with inhalation of H_2_ gas have been conducted in a variety of diseases. For example, H_2_ inhalation improved physical and respiratory function in acute post-COVID-19 patients [30]; a breathlessness, cough, and sputum scale score in patient with acute exacerbation of chronic obstructive pulmonary disease [31]; functional state of red blood cells, which is accompanied by a more favorable course of the early postoperative period [32]; and improved systemic inflammation and liver histology in patients with moderate-severe non-alcoholic fatty liver disease [33].

As the molecular mechanism by which H_2_ exerts multiple functions, a target molecule of H_2_ was recently identified [34]. An oxidized form of porphyrin catalyzes the reaction of H_2_ with hydroxyl radicals, the most oxidative free radicals, to reduce the oxidative stress. Additionally, as a secondary anti-oxidative function, H_2_ activates NF-E2-related factor 2 (Nrf2) [9], which reduces oxidative stress through the expression of a variety of anti-oxidant enzymes [35].

Furthermore, H_2_ has an anti-cell-death function by inhibiting ferroptosis through a decrease in peroxide [36], and by down- and up-regulating pro- and anti-death factors, respectively [37]. 

H_2_ relieves inflammation by decreasing pro-inflammatory cytokines [38]. H_2_ modifies the phospholipid mediator that inhibits Ca^2+^-signaling, resulting in suppressing the nuclear factor of activated T cell (NFAT) transcription pathway to down-regulate pro-inflammatory cytokines [8,36]. NFAT signaling plays an important role in driving amyloid β-mediated neurodegeneration and affects AD [39]. Moreover, the NFAT transcriptional pathway is involved in amyloid β synaptotoxicity [40]. Therefore, the suppression of NFAT transcriptional regulation could explain the beneficial effects of H_2_ for AD improvement.

H_2_ inhibits the free radical chain reaction, resulting in a decrease in fatty acid peroxidation and its end-products such as 4-hydroxyl-nonenal (4-HNE), which is known to serve as a transmitter of various types of cellular signaling. In turn, the decrease in 4-HNE promotes the expression of PGC-1α, followed by increasing FGF21, a key regulator of energy metabolism, and several enzymes related to β-oxidation [41,42]. 

The delivery of drugs to the brain that can cross the BBB is one of the serious issues in developing therapeutic agents for AD. H_2_ is non-polar, non-ionic, and small, allowing it to pass the BBB and easily reach neurons by rapid diffusion [7].

Taken together, given the multiple risks and/or causes for the etiology of AD, H_2_ with these multiple functions is considered to show marked potential to improve AD.

In the present study, we showed that H_2_ treatment led to marked improvement when the effects were compared with those of donepezil: donepezil administration decreased (a lower value means improvement) the ADAS-cog by −2.9 or −3.1 points after 24 weeks [43,44]; however, after 6 months, the scores returned to the initial level despite its continuous administration [21]. On the other hand, in the present study, the mean value of ADAS-cog change was −4.1 points from the initial stage after the 6-month treatment of H_2_. This change by H_2_ may be marked in comparison with approved agents such as donepezil. 

In this follow-up study, the mean of ADAS-cog in the H_2_-treated group remained significantly better than in the untreated control group after 6 months, and the effect tended to persist for 1 year. Moreover, this follow-up in DTI also indicated that the neuronal integrity after 6 months of treatment was significantly higher than at the initial stage. Additionally, the DTI improvement effect tended to persist for one year. These results suggest that 6 months of H_2_ inhalation maintained the disease-modifying effects for at least 6 months. 

Finally, we note the limitations of this study. This study involved a small number of patients enrolled in a non-randomized manner. Further studies require a placebo-controlled, double-blind trial to clarify the effect in a large number of patients. Despite these limitations, this study suggests that H_2_ gas inhalation has the potential to provide not only temporary relief, but also disease-modifying effects.

## 4. Materials and Methods

### 4.1. Approval for This Study

This study was carried out in accordance with “The Code of Ethics of the World Medical Association (Declaration of Helsinki)”. The protocol of this clinical study was approved by the Nishijima Hospital Ethics Committee, and was pre-registered at URL: http://www.jmacct.med.or.jp. Clinical Trial Registration-JMACCT ID: JMA-IIA00308. We received written informed consent from a family member for all patients. 

### 4.2. Patient Selection

The criteria for the inclusion of AD patients were as follows: (1) diagnosis of AD in accordance with the recommendations by the National Institute on Aging-Alzheimer’s Association Group (NIA/AA) [45]; (2) an ADAS-cog score of more than 10 or less than 50 or a corresponding score converted from mini-mental state examination (MMSE) using the formula 70-(MMSE x 2.33) [46]; (3) treatment with at least one of anti-cholinesterase drugs and/or an NMDA receptor antagonist had already been attempted, and yet the ADAS scores were worsening; (4) routine treatment in Neurology Dementia Clinic with multiple ADAS-cog/MMSE tests every 6 months with recent worsening; (5) no significant airway disease such as chronic obstructive pulmonary disease (COPD), pneumonia, bronchitis, or asthma that might interfere with adequate inhalation of H_2_; and (6) having experience of brain MRI. The patients continued to receive at least one of the following medicines: donepezil, galantamine, rivastigmine, or memantine. 

The patients who satisfied these inclusion criteria were offered 1 week of test inhalation and test medication. The patients were confirmed to show no symptoms at a blood level of less than 0.8 mEq/dL, and no kidney or liver dysfunction, and the patients’ families/care givers managed the H_2_ generator including checking the water level of the generator, and let the patients inhale H_2_ gas for 1 h. 

### 4.3. Collection of Data from Untreated Control Patients

As the untreated controls for ADAS-cog assessment, 19 patients with AD out of 94 patients in the Nishijima Hospital database satisfied the following criteria: the initial ADS-cog was in the range from 19 to 40, and the data of the ADAS-cog were followed every 6 months for more than 1.5 years at Nishijima Hospital. Mean and standard deviation values of ADS-cog of the 19 patients were 27.8 ± 3.5 in the initial ADAS-cog, which is similar with that of H_2_ group (Table 1). 

As the untreated control for DTI, 5 patients agreed to the use of their DTI data for this study. Mean ADAS-cog and standard deviation values were 27.6 ± 6.5, which is similar with that of H_2_ group (Table 1).

### 4.4. Treatment 

The patients inhaled 3% H_2_ for 1 h twice daily through a regular facial mask in their own home or in a nursing home. The family members always watched the patients to ensure continuous inhalation for 1 h using the facial mask. H_2_ gas (3%) with 21% oxygen was generated using a portable H_2_ generator (Nishijima/Enoa hydrogen gas generator) as described previously [47]. The H_2_ generator was checked for adequate gas production every month. 

### 4.5. ADAS-cog Examination

Since the Alzheimer’s Disease Assessment Scale-cognitive subscale (ADAS-cog) is widely accepted as one of the most trustworthy methods, we used ADAS-cog as a clinical assessment [20]. In ADAS-cog, a lower value means improvement and a higher value means worse. The ADAS-Cog consist of 11 questions: Word Recall Task, Naming Objects and Fingers, Following Commands, Constructional Praxis, Ideational Praxis, Orientation, Word Recognition Task, Remembering Test Directions, Spoken Language, Comprehension, and Word-Finding Difficulty.

Clinical effectiveness was assessed by monitoring ADAS-cog [20]. ADAS-cog was obtained independently in the physical therapy department, the staff of which did not know whether the subjects were participants in this study or common outpatients. These results were reported to doctors via an electronic chart system in a blinded manner.

### 4.6. Measurement of the Integrity of Neurons by Diffusion Tensor Imaging 

The neuron bundles of cerebral white matter are highly directional. The water diffusivity is much higher along the direction of the alive bundle compared with other directions. The difference between diffusivity along and across bundles increases depending upon neuronal integrity and axonal density. Fractional anisotropy (FA) values reflect the density of active axons within bundles, with lower values corresponding to lower active axonal density. To evaluate the diffusivity of water along the neuron bundles as neuron integrity, diffusion tensor imaging (DTI) was conducted with different FA values at 0.1 and 0.2 for the entire hippocampal region [17,18,19].

Brain MRI was examined in the radiology department as an objective assessment. Five seed points were set at the volumetric measurement sites where the neuronal bundles passed through the entire hippocampus (Figure 2). Digital tractography imaging was performed using Neuro3D with the GRAPPA technique. DTI was obtained with FA values of 0.1 and 0.2. The tract size was calculated from the pixel number of the tract images, and the number of pixels in the tract was calculated using ImageJ software. 

The staff in these departments reported the results to doctors via an electronic chart system in a blinded manner. They had no information on whether subjects were participants in the study or common outpatients.

### 4.7. Statistical Analysis

Statistical analysis was performed by an academic biostatistician using SAS software version 9.2 (SAS Institute Inc., Cary, NC, USA) by the Student’s *t*-test with two tails. *p* < 0.05 is considered as significant.

## 5. Conclusions

This study suggests that H_2_ inhalation resulted in marked improvements as assessed by ADAS-cog and DTI, and importantly, provided not only temporary relief, but also maintained the effect for at least 6 months without H_2_ treatment. However, this study of H_2_ inhalation in AD patients had a small number of patients, and was an open label clinical study. Further studies require a randomized, placebo-controlled group to clarify the effect on a large number of patients. Despite these limitations, we propose that H_2_ inhalation is a candidate for the disease-modifying treatment of AD, as the effects of H2 inhalation is striking.

## 6. Patents

The authors are applicants of a patent that resulted from the work reported in this article. 

## Figures and Tables

**Figure 1 pharmaceuticals-16-00434-f001:**
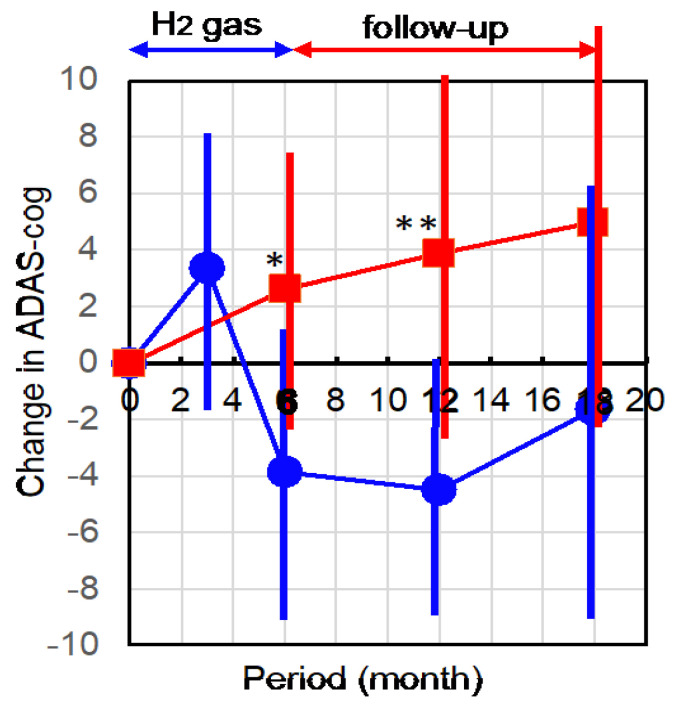
Change of ADAS-cog from the initial stage. Participants inhaled 3% H2 gas for 1 h, twice per day, for 6 months and were followed-up for 1 year without H2 inhalation. ADAS-cog was examined at the first 3 months (only H2 group), and every 6 months in the participants and untreated control patients, and the changes from the initial stage were averaged with standard deviations (error bars). Blue circles and red squares show the H2 group and untreated controls, respectively. * and ** indicate significances with *p* = 0.017 and 0.004, respectively (H2 group vs. untreated control group). Note that ADAS-cog was obtained independently in the physical therapy department, the staff of which did not know whether subjects were participants in this study or common outpatients. These results were reported to doctors via an electronic chart system in a blinded manner.

**Figure 2 pharmaceuticals-16-00434-f002:**
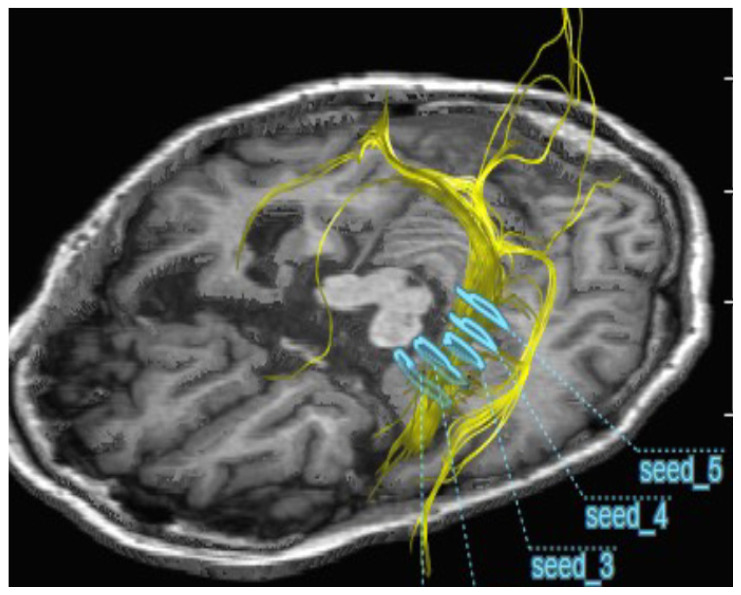
Application of diffusion tensor imaging (DTI) to the neuron bundles that passed through the entire hippocampus. For the selection of neuron bundles that pass through the entire hippocampus, five seed points were set, as shown by blue gates. Yellow neuron bundles that pass through the hippocampus were visualized by the DTI method, as described in Materials and Methods. Three-dimensional images of neuronal bundles in the brain passing through the hippocampus were taken, and two-dimensional images containing the hippocampus were arranged into one image.

**Figure 3 pharmaceuticals-16-00434-f003:**
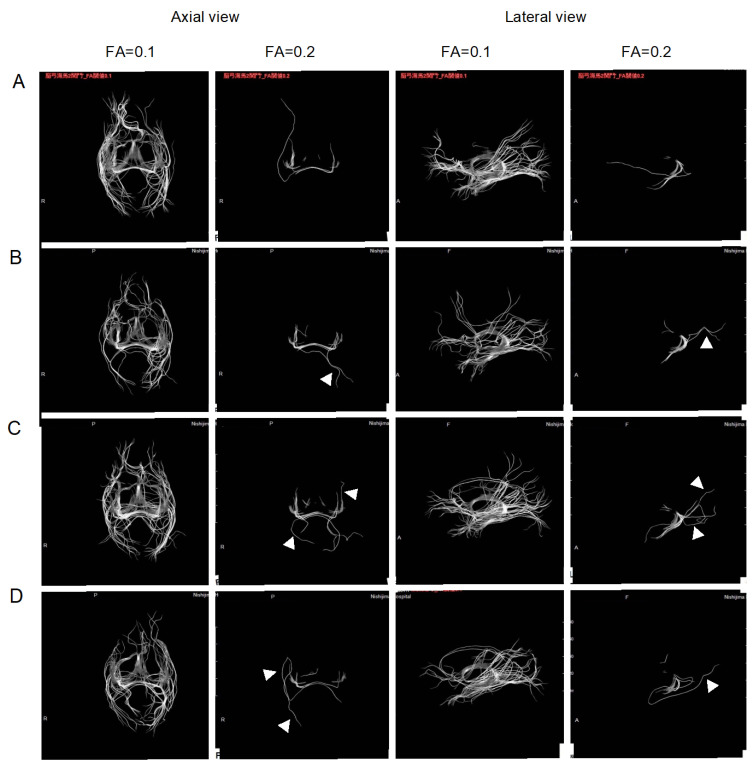
Representative diffusion tensor imaging (DTI) in two patients. The patients inhaled H2 gas for 6 months and were followed-up for 12 months without inhalation of H2 gas. Since DTI changes vary from patient to patient, images of two patients are shown as examples. DTI was visualized at FA = 0.1 and FA = 0.2 with lateral and axial views. (**A**): initial stage, (**B**): after H2 inhalation for 6 months, (**C**): follow-up after the subsequent 6 months, and (**D**): follow-up after 12 months. Arrowheads indicate the area that was changed in shape.

**Figure 4 pharmaceuticals-16-00434-f004:**
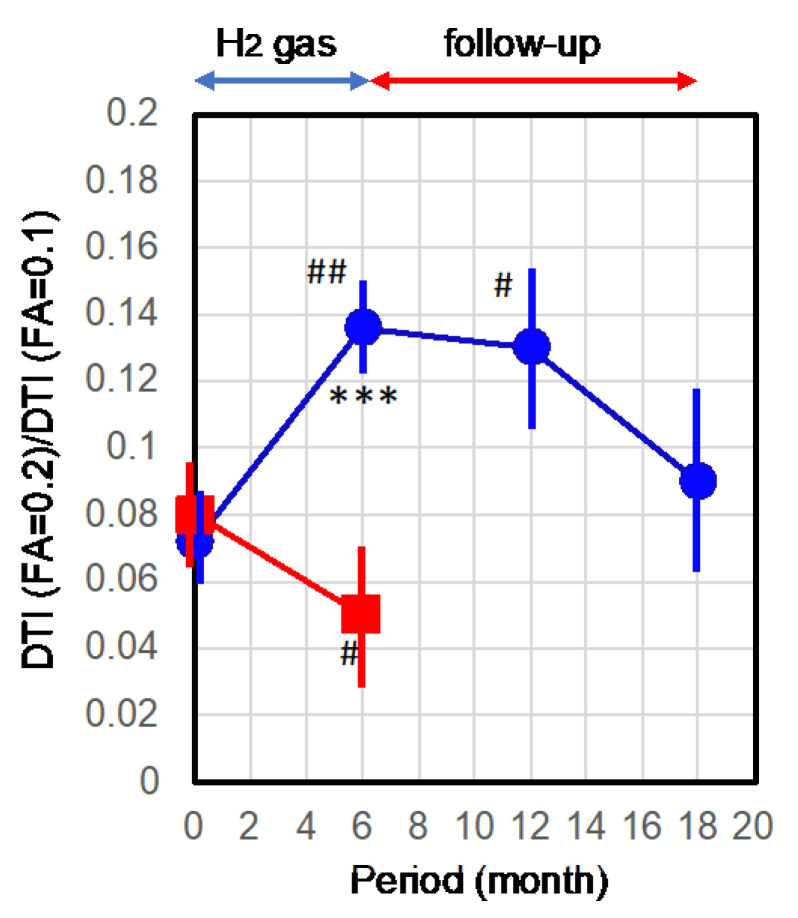
Quantitative analyses of the time course of the diffusion tensor imaging (DTI). Participants inhaled H2 gas for 6 months and were followed-up for 12 months without inhalation of H2 gas. DTI was examined at the indicated times for participants and untreated control patients. The pixel numbers of the corresponding neuronal bundle tracts in FA = 0.1 and FA = 0.2 were obtained for the right and left hemispheres with lateral and axial views. Each value in FA = 0.2 was divided by that in FA = 0.1, and these normalized values from patients with 4 images were averaged with the standard errors (error bars). Blue circles and red squares show the H2 group and untreated controls, respectively. *** indicates significance with *p* = 0.001 vs. untreated group. ## and # in the H2 group indicate significances with *p* = 0.0036, and *p* = 0.011, respectively, vs. the initial stage of the H2 group. # in the untreated control group indicates significance with *p* = 0.024 vs. the initial stage of the control group.

**Table 1 pharmaceuticals-16-00434-t001:** Background of the patients with AD.

	H_2_-Treated Group		Control for DTI		Control for ADAS-cog	
Number of patients	8		5		19	
Age (±SD)	79.4	±6.11	80.4	±1.8	-	-
Creatine (mg/dL) (±SD)	0.675	±0.104	0.66	±0.114	-	-
Female (%)	87.5		80		-	-
Diabetes (%)	12.5		20		-	-
Lipidemia (%)	25		60		-	-
Mean of ADAS-cog (±SD)	29.74	±8.03	27.6	±6.5	27.83	±3.53

SD: standard deviation. -: data not available.

## Data Availability

Data is contained within the article.

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
