# Peer review of "Therapeutic Inhalation of Hydrogen Gas for Alzheimer’s Disease Patients and Subsequent Long-Term Follow-Up as a Disease-Modifying Treatment: An Open Label Pilot Study"

_pharmaceuticals, 2023, doi:10.3390/ph16030434_

Round 1
Reviewer 1 Report
The work by Ono et al, is an open label clinical study on the potential use of H2 as a therapeutic agent for Alzheimer Disease.
The topic of the study is very interesting, and the results provided are promising, however the data could be presented in a better way.
For example, Figure 2 is not clear at all. Why there are yellow fibers shown outside the brain? there is also a name-SUZUKI Y at the top of the panel. Is that the name of a patient? or a doctor? It should be removed.
There are two different Figures shown with the same label, Figure 3, but they do look different why?
And the legend does not provide any information on the panels shown.
Moreover there are a lot of spelling mistakes throughout the manuscript. These should be fixed.
Author Response
To Reviewer 1
Thank you very much for your valuable comments. According to the suggestion, we revised the manuscript. Point-to-point responses are as follows.
Comment. Figure 2 is not clear at all. Why there are yellow fibers shown outside the brain?
Response: We agree with the suggestion. We added the following sentence to help understandings of Figure 2. “Three-dimensional images of neuronal bundles in the brain passing through the hippocampus were taken, and two-dimensional images containing the hippocampus were arranged into one image” (Line 165).
Comment: there is also a name-SUZUKI Y at the top of the panel. Is that the name of a patient? or a doctor? It should be removed.
Response: According to the suggestion, we removed the region suggesting the name in Figure 3.
Comment: There are two different Figures shown with the same label, Figure 3, but they do look different why? And the legend does not provide any information on the panels shown.
Response: I agree with the suggestion. We explained that two sets of panels were obtained from two patients and the reason as “Since DTI changes vary from patient to patient, images of two patients are shown as examples” (Line 293 ).
Reviewer 2 Report
Line 201: How data was normalized?
208: In “follow-up study without H2 gas inhalation”, “without inhalation” should be removed because it can indicate incorrectly that this group did not inhale H2 gas. The use of “treated group” could be a good alternative.
Author Response
To Reviewer 2
Thank you very much for your valuable comments. According to the suggestions, we revised the manuscript. Post-to-point responses are as follows.
Comment: How data was normalized?
Response: We explained why the normalization is necessary and how the normalization was performed in the text and the figure 4 legend as “Values for each measurement variation should be normalized for more accurate quantitative analysis. For this purpose, the value of FA=0.2 was divided by the value of FA=0.1 obtained at the same time and corrected for variation between measurements for normalization. Therefore, the pixel numbers of the corresponding neuronal bundle tracts in FA=0.1 and FA=0.2 were obtained for the right and left hemispheres with lateral and axial views. Each value in FA=0.2 was normalized with that in FA=0.1, and the normalized values from patients with 4 images were averaged with the standard errors (error bars)” (Line 320).
Comment: In “follow-up study without H2 gas inhalation”, “without inhalation” should be removed because it can indicate incorrectly that this group did not inhale H2 gas. The use of “treated group” could be a good alternative.
Response: According to the suggestion, we removed “without inhalation” and revised the sentence to “In this follow-up study, the mean of ADAS-cog in the treated group remained significantly better than that in the untreated control group after 6 months” (Line 371).